# Nutritional Composition of *Apis mellifera* Drones from Korea and Denmark as a Potential Sustainable Alternative Food Source: Comparison Between Developmental Stages

**DOI:** 10.3390/foods9040389

**Published:** 2020-03-27

**Authors:** Sampat Ghosh, Ho-Yong Sohn, Su-Jin Pyo, Annette Bruun Jensen, Victor Benno Meyer-Rochow, Chuleui Jung

**Affiliations:** 1Agricultural Science and Technology Research Institute, Andong National University, Andong 36729, Korea; sampatghosh.bee@gmail.com (S.G.); meyrow@gmail.com (V.B.M.-R.); 2Department of Food and Nutrition, Andong National University, Andong 36729, Korea; hysohn@anu.ac.kr (H.-Y.S.); vytn0608@naver.com (S.-J.P.); 3Department of Plant and Environmental Science, University of Copenhagen, 1871 Frederiksberg, Denmark; abj@plen.ku.dk; 4Department of Genetics and Ecology, Oulu University, SF-90140 Oulu, Finland; 5Department of Plant Medicals, Andong National University, Andong 36729, Korea

**Keywords:** Amino acids, fatty acids, minerals, antioxidant, antimicrobial, supplement, sustainable food, food safety

## Abstract

We compared nutrient compositions of honey bee (*Apis mellifera)* drones of different developmental stages from two different populations—the Italian honey bee reared in Korea and Buckfast bees from Denmark. Analyses included amino acid, fatty acid, and mineral content as well as evaluations of antioxidant properties and haemolysis activities. The compositions of total amino acids, and thus protein content of the insects, increased with development. A similar trend was observed for minerals presumably due to the consumption of food in the adult stage. In contrast, total fatty acid amounts decreased with development. Altogether, seventeen amino acids, including all the essential ones, except tryptophan, were determined. Saturated fatty acids dominated over monounsaturated fatty acids in the pupae, but the reverse held true for the adults. Drones were found to be rich in minerals and the particularly high iron as well as K/Na ratio was indicative of the nutritional value of these insects. Among the three developmental stages, adult Buckfast drones exhibited the highest antioxidant activity. Bearing in mind the overall high nutritional value, i.e., high amino acids, minerals and less fatty acids, late pupae and adult drones can be useful for human consumption while the larvae or early pupal stage can be recommended as feed. However, owing to their relatively high haemolysis activity, we advocate processing prior to the consumption of these insects.

## 1. Introduction

In the past few decades, following an earlier suggestion by Meyer-Rochow [1] to consider insects in combatting global nutritional impasses, insects have received significant attention as a potential sustainable food item for humans. Entomophagy, i.e., the practice of eating insects [2], has existed and still exists as a cultural attribute in many different traditional communities around the globe [3,4,5,6,7,8,9,10,11]. 

Scientific reports, possibly exaggerated, claim that almost 2 billion people eat insects regularly as a planned part of their diet [12], but the total number of insects considered edible around the world is almost certainly more than the 2000 mentioned by Jongema [13]. Priority has been given to insects primarily because of their rich nutrient composition and sustainability—insects require fewer resources to rear and have lower environmental impacts than conventional livestock [12,14]. However, the domestication of insect species appears not to have been easy or much needed, judging by the small number of truly domesticated species like silkworms and honey bees. Honey bee domestication started 3–5000 years ago in the civilizations of the Middle East. The nutritional as well as therapeutic value of honey has been appreciated and mentioned in ancient texts like Ayurveda, the Bible and the Quran. Beeswax, royal jelly, bee pollen, propolis and bee venom are further significant products of bees and even the word “medicine” has its origin from the fermented honey known to Anglo-Saxon tribals as ‘mead’. 

What is, however, less appreciated is that the bees themselves and not just their honey is nutritious. To cite some examples, Australian Aboriginal people consume the honey plus specimens of native bees as food [15]; the Hazda people of Tanzania do not remove bee larvae from the combs that they eat [16] and in Thailand more than 10,000 colonies of *Apis florea* are annually harvested by ‘bee hunters’ as people consume larvae, pupae and the honey together [17]. Use of honey bee brood as food is also common in Mexico [18], Ecuador [19], Zambia [20], Senegal [21,22], and China [23]. 

A sensory analysis of drone brood found developmental stages, worker larvae and pupae to possess differences in their sensory profiles [24]. Nutrient composition and values of different developmental stages and honey bee species like *Apis mellifera*, *A. cerana*, *A. dorsata* have been studied [25,26,27], but these studies focused on worker brood, whereas the drones remained unobserved. In contrast to female queen and worker honey bees, drones are produced parthenogenetically principally by queen bees and to a minor extent by workers, and they are thus haploid [28]. 

The major task of the drones is to mate with virgin queens and to assure genetic mixing and natural reproduction. The bigger bodies of drone brood in comparison with worker bees likely makes them more susceptible to the attack by *Varroa* mites and removing drone broods from the hive as a way to control *Varroa* mite infections is a common practice in many countries [29]. Thus, the drone brood remains unused (despite a call to use them as food by Ambühl [30], who also provided recipes to prepare them). It makes sense, both ecologically and economically, to use drone brood, a by-product of the hive [22,31], as human food or animal feed. The usage of drone bees could help enhance the economic condition especially of the small- to medium-scale beekeepers. However, reports of the nutrient composition of honey bee drones are limited. It therefore seemed important to examine and put on record the chemical composition, including amino and fatty acids as well as mineral content of drones, their larvae and pupae to gauge the nutritional value of the different developmental stages. The data would then allow us to draw conclusions on likely functional consequences of drone consumption. The study’s aim was further to clarify if there were any differences in the body composition of honey bee drones from two locations of the northern hemisphere separated by 8500 km and 20 degrees of latitude.

## 2. Materials and Methods

### 2.1. Nutritional Composition of Drones of Italian Bees and Buckfast Bees 

#### 2.1.1. Sample Preparation

Drone larvae, pupae and adults of the honey bee *Apis mellifera* were obtained from two different countries in two different continents separated by 8500 km and 20 degrees of latitude (*n* = 20 for each studied developmental stage of the drones)—Buckfast bees from Europe (Denmark) and Italian bees from Asia (South Korea). The Buckfast bee is a hybrid bee bred according to the Buckfast system, which is a mixture of many *A. mellifera* subspecies, primarily *Apis mellifera ligustica* and *A. m. mellifera*. In Denmark, drones of the Buckfast bee were collected from a healthy colony in the apiary located at the University of Copenhagen, Frederiksberg Campus, during the late summer of 2016. Drones of the Italian bees were obtained from a healthy colony of the experimental apiary of Andong National University, South Korea during the autumn of 2019. The samples were categorized based on developmental stage—larvae (fifth larval instars from the open-cell phase), early pupae (when eye colour was not yet present), late pupae (when dark eye colour was apparent), young adults (just emerging) and old adults >7 days. Wings of the adult drones were removed. All the samples were freeze-dried for at least 72 h at −50 °C (Alpha 1-4 LD Plus, Christ, Osterode am Harz, Niedersachsen, Germany). The dried samples were ground into powder and stored at −20 °C until further analyses. All glassware used for the chemical analyses was meticulously cleaned and the chemicals used for the analytical purposes were of pure HPLC grade. Weights of the different developmental stages of the Italian bee drones were taken with the help of Acculab weighing equipment (ALC310.3, Kern, Kingswinford, West Midlands, UK) with a precision of 0.001 g. However, Buckfast drones were received in a freeze-dried condition from Denmark, which is why their fresh weights could not be recorded.

#### 2.1.2. Amino Acid Analysis

The amino acid composition was estimated using a Sykam Amino Acid analyzer S433 (Sykam GmbH, Eresing, Bavaria, Germany) equipped with Sykam LCA L-07 column following the standard method [32]. The samples (20 mg) were hydrolyzed in 6 N HCl for 24 h at 110 °C under a nitrogen atmosphere and then concentrated in a rotary evaporator. The concentrated samples were reconstituted with sample dilution buffer supplied by the manufacturer (physiological buffer 0.12N citrate buffer, pH 2.20). The hydrolyzed samples were analyzed for amino acid composition. 

#### 2.1.3. Fatty Acid Analysis

Fatty acid composition was analyzed following the standard method of the Korean Food Standard Codex [33] using gas chromatography-flame ionization detection (GC-14B, Shimadzu, Tokyo, Japan) equipped with an SP-2560 column. The samples were derivatized into fatty acid methyl esters (FAMEs). Identification and quantification of FAMEs were accomplished by comparing the retention times of peaks with those of pure standards purchased from Sigma (Yongin, Republic of Korea) and analyzed under the same conditions.

#### 2.1.4. Mineral Analysis

Minerals were analyzed following standard procedures according to the Korean Food Standard Codex [33]. Samples were digested with nitric and hydrochloric acid (1:3) at 200 °C for 30 min. Each sample was then filtered using Whatman filter paper (0.45 micron) and stored in washed glass vials before analyses could commence. The mineral contents were analyzed using an inductively coupled plasma-optical emission spectrophotometer (ICP-OES 720 series; Agilent; Santa Clara, CA, USA).

### 2.2. Functional Properties of Buckfast Drone Bee Ethanol Extracts

#### 2.2.1. Sample Preparation

To prepare ethanol extracts, 10% (w/v) of bee larva, pupa and adult lyophilized powders was dissolved in 95% ethanol by powder weight (Daejung Chemicals & Metals Co., Ltd., Siheung, Korea) and extracted three times at room temperature. After that, the extract was paper-filtered (Whatman No. 2) and concentrated under reduced pressure (Eyela Rotary evaporator, N-1000, Tokyo Rikakikai Co., Ltd, Tokyo, Japan). A freeze dryer (FD5508, Ilshin Lab Co. Ltd, Dongduchun, Korea) was used to prepare the powder. Extracted samples were dissolved in DMSO at a suitable concentration, for functional component analysis. Functional analyses were restricted only to Buckfast drone because of the limited accessibility. We therefore carried out the examination of functional properties.

#### 2.2.2. Total Polyphenol, Flavonoid, Reducing Sugar

Total polyphenol content in bee larvae, pupae and adults was measured colorimetrically by using Folin–Ciocalteu’s phenol reagent [34]. Distilled water of 2.6 mL and 200 µL of Folin-Ciocalteu’s phenol reagent added to 200 µL of the sample were mixed together; the mixture was allowed to react for 6 min at room temperature and then 2 mL of 7% (w/v) Na_2_CO_3_ solution were added. The mixture was subsequently allowed to react for 30 min at 30 °C and the absorbance was measured using a spectrophotometer (Epouch Microplate reader, Biotek Instrument Inc., Winooski, VT, USA) at 700 nm. A standard curve was constructed using rutin as the standard. 

The total flavonoid content of bee larvae, pupae and adults was measured by the aluminium colorimetric method [35]. Distilled water of 320 µL and 15 µL of 5% (w/v) NaNO_2_ to 100 µL of the sample were mixed together and allowed to react for 5 min. Then, 10% AlCl_3_ solution was added and the mixture was reacted for 1 min, before 1M NaOH was added. Subsequently, the absorbance was measured at 510 nm. A standard curve was constructed using tannic acid as a standard substance. Total sugar content was determined by the phenol-sulphuric acid method in microplate format [36] and the reducing sugar content was measured by the DNS method [37]. Standard curves were constructed using sucrose and glucose, respectively. 

#### 2.2.3. Antioxidant Activity

DPPH (1,1-diphenyl-2-picryl hydrazyl) anion scavenging ability, ABTS (2,2-azobis(3-ethylbenzothiazoline-6-sulfonate)) cation scavenging activity, nitrite scavenging activity and reducing power were evaluated following previously reported methods [38]. Vitamin C (Sigma Co., St. Louis, MO, USA) was used as a positive control.

#### 2.2.4. Antimicrobial Activity

The bacteria used were Listeria monocytogenes KACC 10550, Staphylococcus epidermidis ATCC 12228, Staphylococcus aureus KCTC 1916, Bacillus subtilis KCTC 1924 for Gram-positive, and Escherichia coli KCTC 1682, Pseudomonas aeruginosa KACC 10186, Salmonella typhimurium KCTC 1926 and Proteus vulgaris KCTC 2433 for Gram-negative bacteria) cultured in nutrient broth (Difco Co., Sparks, MD, USA) at 37 °C for 24 h. Yeasts like Candida albicans KCTC 1940 and Saccharomyces cerevisiae IF0 0233 were cultured in Sabouraud dextrose agar (Difco Co. Sparks, MD, USA) at 30 °C for 24 h. Antimicrobial activity was determined by the disc diffusion method [39]. A volume of 100 µL of each preculture was spread on agar plates; filter paper (Whatman No.2) soaked with 5 µL of bee extracts was loaded on the agar plate. After 24 h of culture, the size of the growth inhibition zone was measured. Ampicillin and miconazole were used as positive controls to be compared with the ethanol extracts of the bee samples. 

#### 2.2.5. Haemolysis Activity

A human blood sample was diluted in phosphate-buffered saline (PBS; pH 7.4) and centrifuged at 200× *g* for 10 min. After three washes, the final concentration of the erythrocytes was adjusted to 4%. The erythrocyte suspension was transferred into 96-well plates and incubated with the extracts of the bee larvae, pupae and adults at 37 °C for 1 h. The plate was centrifuged at 200× *g* for 10 min. The supernatant was collected and aliquoted, and haemolysis was evaluated by determining the release of haemoglobin from the 4% human erythrocyte suspension at 414 nm with an ELISA reader. The 0% and 100% haemolysis activities were evaluated with PBS alone and 0.1% Triton X-100, respectively. Haemolysis percentage was calculated using the following equation—haemolysis (%) = ((*A*_414 nm_ in the bee samples − *A*_414 nm_ in PBS)/(*A*_414 nm_ in 0.1% Triton X-100 − *A*_414 nm_ in PBS)) × 100 [40].

### 2.3. Statistical Analysis

In order to increase reliability, all the chemical analyses were performed in triplicate and expressed as mean ± SD except for the case of fatty acid and mineral analysis. For body weight, *n* = 9 individuals were taken for late pupae, early adult and adult Italian bee drones, and *n* = 5 individuals were taken for early pupae. The composite sampling technique has been followed for sampling for each experiment. To test the differences, in case of the drone’s weight and the functional properties of Buckfast drones, one-way analysis of variance (ANOVA) followed by Posthoc test (LSD) were carried out using SPSS 16.0. If the *p* value e was found ≤0.05 (CI = 95%), the null hypothesis was rejected.

## 3. Results and Discussion

### 3.1. Body Weight of Different Developmental Stages of Drone

Body weights of Italian honey bee drones were found to differ significant between pupae and adults, ranging from 0.34 to 0.27 g (ANOVA, *p* < 0.05) (Figure 1). Early pupae (0.34 ± 0.02 g) were found to have a higher body weight than late pupae (0.30 ± 0.02 g) and early adults (0.27 ± 0.01 g). However, the body weight then increased again as the adults aged. After emergence, drones are initially fed by workers. The former then commence reproductive preparations for their mating flights, which would take place after 7–12 days following eclosion [41].

### 3.2. Nutritional Composition of Drone Bees

#### 3.2.1. Amino Acid Composition

The amino acid composition of different developmental stages of drones is shown in Table 1. Altogether, seventeen amino acids were found. A significantly increasing amount of amino acid content (and therefore protein) was found to be associated with the developmental stage of a drone (ANOVA, *p* < 0.05 for Italian bees and Buckfast bees). This was in agreement with previous reports on worker honey bees [26,27]. Assuming the total amino acid content reflects protein content of a drone, the value was higher or comparable to the protein content of conventional foods of animal origin (pork: 27.7%, beef: 40.5%, chicken: 54.7%, egg: 52.7%), i.e., values for conventional food obtained from the USDA database and calculated on dry matter basis. Glutamic acid was the most abundant and very important amino acid as it is the precursor of γ-amino butyric acid (GABA), an inhibitory neurotransmitter. Leucine followed by lysine was dominant among the essential amino acids. A similar pattern of amino acid distribution was reported by Kim et al. [42], but methodological differences to our study existed. Lysine receives attention as it is a limiting amino acid in many cereal-based diets [43,44], which are most prevalent in developing countries. Lysine is the precursor of carnitine, which plays a critical role in the carrying process of fatty acids [45].

Phenylalanine and tyrosine together were present in high amounts. Phenylalanine is the precursor of tyrosine, which synthesizes dopamine, adrenaline (epinephrine) and noradrenaline (norepinephrine) [46]. Tryptophan was not determined and cysteine and methionine were not recovered entirely, presumably because of the incomplete acid hydrolysis of the sample and the relatively small amounts of these amino acids [47]. However, the presence of sizable amounts of essential amino acids increases the food value of drone brood. Valine, leucine and isoleucine are three branched-chain amino acids playing essential roles in the maintenance of muscle tissue. Threonine is an important component of structural proteins. Moreover, it plays a critical role in the maintenance of the intestinal mucosal integrity and barrier function [48]. Histidine is the precursor of histamine, which plays a vital role in immune responses [49]. Methionine plays important roles in metabolic affairs and detoxification. 

The amino acid contents measured were within the range of previously reported values for edible insects belonging to different orders [50,51,52,53,54,55]. Adult Buckfast honey bee drones contained less total amino acid than adult *A. mellifera ligustica* drones did. The differences between the amino acid content are likely to be due to the different geographical locations and environmental conditions as well as the locally available pollen resources for the bees’ nutrition. The ages of the adults and their physiologic adaptations are additional factors to be considered. Younger drones are capable of synthesizing some amino acids like proline more successfully than the older adults do [56]. However, it was not possible to categorize the adult drone population samples according to age while estimating their nutritional composition. 

#### 3.2.2. Fatty Acid Composition

Table 2 represents the fatty acid compositions of different developmental stages of drone bees. A gradual decreasing trend, although not statistically examined, with developmental stage was found for total fatty acid content (Table 2); the observation being relevant with regard to previous reports on worker bees of *A. mellifera* and *A. cerana* [26,27]. SFA (saturated fatty acid) and MUFA (monounsaturated fatty acid) proportions sharply decreased with the progression of the developmental stages, while in contrast, the proportion of PUFAs (polyunsaturated fatty acids) increased. In all cases the proportion of MUFAs dominated. Palmitic acid followed by stearic acid was the most abundant saturated fatty acids. On the other hand, oleic and α-linolenic acid dominated among the MUFAs and PUFAs, respectively. 

From a nutritional point of view, SFAs, particularly lauric, myristic and palmitic acid, but not stearic acid, are deemed to be undesirable as they are thought to increase body cholesterol [57]. All of the three undesirable fatty acids were decreasing as the drones got older. In almost all of the cases, only very small amounts of PUFAs were found, a finding that agrees with results obtained earlier from worker honey bees like *A. mellifera*, *A. cerana* and *A. dorsata* workers [26,27]. Linoleic and α-linolenic acids are essential for most insects including honey bees, although some insect species are capable of synthesizing linoleic acid [58]. Being the major source of ω-3 fatty acids, α-linolenic acid plays important roles in the context of reproduction. Studies by Ma et al. [59] have revealed that α-linolenic acid provided with a 2% to 4% pollen substitute was found to be optimal for the highest reproductive demands while deficiencies of ω-3 fatty acids were found to impair learning in honey bees [60]. The higher MUFA content of drones rather than worker bees is advantageous from the standpoint of human nutrition as it reduces low density lipoprotein (LDL) [61,62]. 

#### 3.2.3. Mineral Content

Table 3 represents the content of nutritionally important minerals of the different developmental stages of drone bees. In general, adult drones contained quantitatively higher amounts of minerals than those reported for the brood, although not statistically examined. The most abundant mineral was phosphorus followed by potassium (Table 3). Phosphorus is essential for ATP and energy production. Phosphorus is also required to form mineral salts with calcium that play critical roles in the formation of bones and teeth. Calcium plays important roles in several other physiological functions including muscle contraction and relaxation, nerve functioning, blood clotting, etc. The most common food sources of calcium are dairy products and meat, items which are often unaffordable for people in developing and underdeveloped regions of the world. High potassium and a high K/Na ratio are advantageous for human nutrition particularly for those sections of the population suffering from hypertension [63,64]. Sodium and potassium are required for the appropriate maintenance of electrolyte and proper fluid balance, muscle contraction and nerve transmission. The sodium content of Buckfast honey bee drones from Denmark was much higher than that of Italian honey bee drones from Korea. In contrast, the copper content of Buckfast honey bee drones from Denmark was less than that of Italian honey bee drones from Korea. Differences in the mineral contents are presumably due to different resources available for the bees’ nourishment in the two different geographical locations. Iron is a mineral of interest as its deficiency often leads to anaemia and because the most vulnerable populations for iron deficiencies are underprivileged women and children in the poorest regions of the world [65,66]. Adult drones contained higher amounts of iron than their pupae. Assuming good bioavailability, it can be expected that a diet supplemented with drones can help ameliorate iron deficiencies. Zinc, another essential mineral, plays critical roles in many metabolic pathways including DNA replication, transcription and protein synthesis [67], and the relatively high amounts of this element in drones has to be seen as a bonus in connection with drones as part of the human nutrition. 

### 3.3. Functional Properties of Buckfast Honey Bee Drone Bee Ethanol Extract

#### 3.3.1. Total Polyphenol, Flavonoids, Reducing Sugar Content

Figure 2 represents total polyphenol, total flavonoid, total sugar and reducing sugar content of different developmental stages of drones. The total polyphenol content of adult drones (12.4 ± 0.2 mg/g) was found to be higher than that of the brood stages (6.8 ± 0.9 mg/g for larvae, and 5.6 ± 0.2 mg/g for pupae). The polyphenol content of the brood stage of the drone was comparable to the value reported for tannin content of edible insects like *Oecophylla smaragdina* (4.97 ± 0.5 mg/g) and *Odontotermes* sp. (6.15 ± 0.6 mg/g) [52]. Adeduntan reported tannin content of edible herbivorous insects from Nigeria within the range of 2.5 to 11.5 mg/g [68]. No significant differences were found in regard to flavonoid content of different developmental stages (5.8 ± 1.4, 4.7 ± 0.9 and 4.5 ± 0.3 mg/g for larvae, pupae, and adults, respectively). Total sugar and reducing sugars were found to be more abundant in the adults (56.7 ± 1.5 mg/g) than other developmental stages (2.7 ± 0.1 mg/g for larvae and 2.1 ± 0.2 mg/g for pupae). Total sugar and reducing sugar content of adult drones was found to be much higher than that reported for five edible insects like *Allomyrina dichotoma*, *Tenebrio molitor*, *Protaetia brevitarsis*, *Gryllus bimaculatus* and *Teleogyllus emma* [69]. Reactive oxygen species (ROS), including hydrogen peroxide, hypochlorous acid and free radicals such as hydroxyl radical and superoxide anions, are readily produced by several oxidation processes in the human body as part of the metabolism. These ROS are often potentially damaging to cells and prolonged oxidative stress has even been associated with the risk of developing diabetes, heart disease, cancer and aging processes [70]. To balance the oxidative stress, animals including humans have developed a complex system of antioxidants such as glutathione and enzymes like catalase and superoxide dismutase. However, dietary supplements of antioxidants are also immensely important to balance oxidative stresses. Polyphenols are an important group of compounds showing antioxidant activity, which is due primarily to resonance stabilization of the phenoxyl radical after oxidation [71]. 

Flavonoids are phenolic compounds, also exhibiting antioxidant activity [72]. To cite but a few examples, flavonoids inhibit enzymes responsible for superoxide anion production such as xanthin oxidase, protein kinase, etc. [73,74], and a number of flavonoids have been shown chelating trace metals, which play critical roles in enhancing ROS formation [72]. Flavonoids also act as antimicrobials, photoreceptors, visual attractors and feeding repellents of herbivores in plants [72]. Reducing sugars are capable of acting as reducing agents since they contain free aldehyde or ketone groups. Excessive sugar consumption is one factor promoting an overweight status and obesity leading to diabetes [75]. In contrast, an intake of reducing sugars lowers the risk of becoming overweight or obese. Thus, the consumption of drones, including the adults, should not increase the risk of obesity and associated complications. 

#### 3.3.2. Antioxidant and Antimicrobial Profiles 

Table 4 represents antioxidant activity, including DPPH, ABTS, nitrite and reduction potential of drone larvae, pupae and adults. Adult drones exhibited the highest antioxidant activity followed by pupae and larvae, which is presumably due to the presence of high amounts of polyphenols. Furthermore, high amounts of amino acids like aspartic acid, glutamic acid, cysteine and lysine can improve the consequences of oxidative stress [76,77]. The gradual increase in their amount along with the developmental stages of the drones can justify the higher antioxidant activity of adults. Table 5 represents the antimicrobial activities of the different developmental stages. No antibacterial and antifungal activities were noticed in the present study. 

#### 3.3.3. Haemolysis Activity 

Figure 3 represents the haemolytic activity of drone larvae, pupae and adults. Ethanolic extraction of drone powder exhibited dose-dependent haemolysis activity. All three different stages showed high haemolytic activity at a concentration of 1000 µg/mL, which is comparable to the antifungal drug Amphotericin B at a concentration of 100 µg/mL. The haemolytic activity shown by the drone brood and adults was found to be much higher than that reported for other studied edible insects (communicated 69) as well as many medicinal plants [78,79]. Out of 222 plants, Hossaini, for example, discovered 20 species that exhibit haemolytic activities [80]. However, a detailed study on the mechanism of the observed haemolysis and the active compounds responsible for the haemolytic activity is yet to be carried out. Therefore, based on the present results we do not advocate the raw consumption of drone bees, but recommend that the drone brood and adults be processed. This could involve blanching, drying, boiling, etc., and should lead to increased safety [81].

## 4. Conclusions

Drone brood (especially at the late pupal stage and as adults) can be recommended as an alternative and nutritious food item for humans as it contains almost all the essential amino acids needed by humans, as also proposed by Ulmer et al. [82]. Similarities in the nutrient content between Italian and Buckfast drones especially with regard to amino and fatty acids were apparent and most likely represent a reflection of the physiological processes shared by the two breeds. On the other hand, discrepancies in the values of some minerals like sodium and potassium in adult populations of both drones very likely stem from distinct environments and their ecological characteristics as well as the different food plants used by the bees. From a nutritional standpoint, the relatively small amounts of fatty acids and the richness in minerals as well as antioxidant activities would provide an extra advantage. 

On the other hand, the larvae or the early pupal stage, with less total amino acid and, thus, protein, than further developed stages, can be recommended as animal feed. However, the examination of the functional properties of Italian drones remains a task for future investigation. Scientific studies showed the efficacy of drone brood, although not at the exact stage mentioned, in terms of improvement of reproductive quality of pigs [83]. However, marketing of drone brood as food per se or a food supplement requires legislation from governmental agencies and permits for the commercialization of this insect. Obviously, a standard dossier of beekeeping practices is essential to ensure the hygiene of the honey bee drone production and to address potential safety issues. Once these obstacles are overcome, the use of drones as food or feed can be a win-win solution for bee keepers and as well as the consumers. 

## Figures and Tables

**Figure 1 foods-09-00389-f001:**
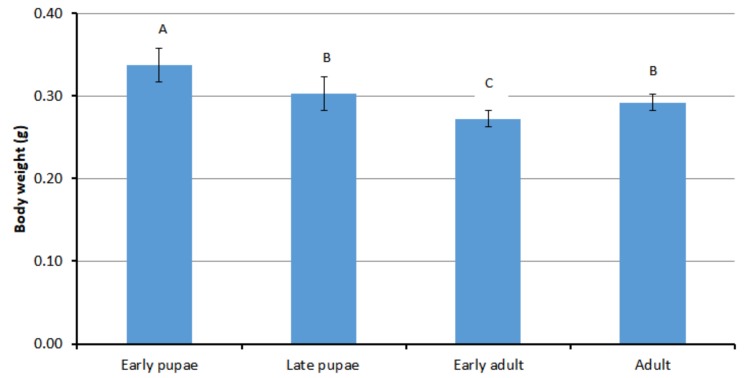
Body weights of different developmental stages of Italian honey bee drones from Korea. Different superscripts indicate statistically significant differences (*p* < 0.05).

**Figure 2 foods-09-00389-f002:**
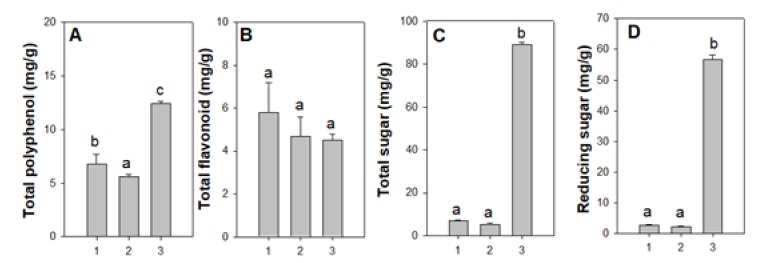
Functional component analyses of ethanol extracts of Buckfast honey bee drone larvae, pupae and adults from Denmark; **A**: Total polyphenol (mg/g), **B**: Total flovaonoid (mg/g), **C**: Total sugar (mg/g), and **D**: Reducing sugar (mg/g). Numbers 1, 2, 3 represent larvae, late pupae and adults, respectively. Different superscripts indicate statistically significant differences (*p* < 0.05).

**Figure 3 foods-09-00389-f003:**
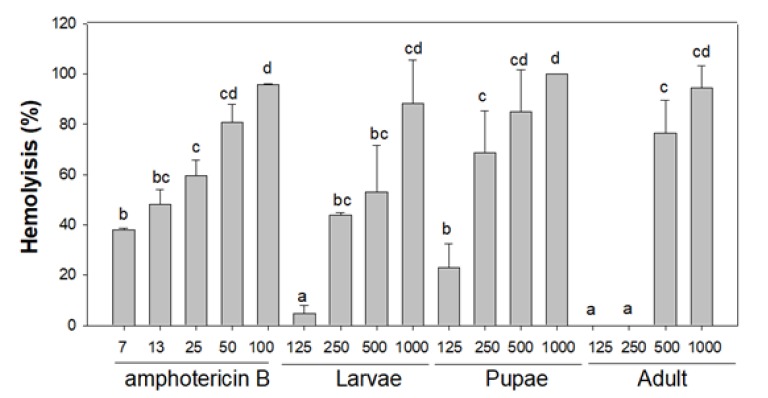
Haemolytic activities of the ethanol extracts of Buckfast honey bee drone; larvae, late pupae and adults from Denmark. Amphotericin B was used as positive control. Different superscripts indicate statistically significant differences among treatment concentrations (*p* < 0.05).

**Table 1 foods-09-00389-t001:** Amino acid composition (g/100 g dry matter basis) of different developmental stages of Italian honey bee drones from Korea (K-) and Buckfast honey bee drones from Denmark (D-).

Amino Acid	D-Larvae	D-Late Pupae	D-Adult	K-Early Pupae	K-Late Pupae	K-Early Adult	K-Adult
Valine *	2.87 ± 0.099	2.97 ± 0.198	3.79 ± 0.480	2.56 ± 0.016	2.97 ± 0.030	4.07 ± 0.043	4.22 ± 0.022
Isoleucine *	2.43 ± 0.021	2.56 ± 0.014	3.28 ± 0.340	2.13 ± 0.004	2.44 ± 0.014	3.16 ± 0.025	3.27 ± 0.007
Leucine *	3.96 ± 0.014	4.26 ± 0.092	5.51 ± 0.062	3.54 ± 0.004	4.14 ± 0.036	5.53 ± 0.043	5.65 ± 0.010
Lysine *	3.52 ± 0.035	3.68 ± 0.021	4.35 ± 0.521	3.00 ± 0.007	3.51 ± 0.023	4.43 ± 0.004	4.56 ± 0.009
Tyrosine **	2.55 ± 0.042	2.76 ± 0.007	2.77 ± 0.219	2.20 ± 0.003	2.77 ± 0.038	3.04 ± 0.019	2.87 ± 0.053
Threonine *	1.86 ± 0.035	1.57 ± 0.134	1.95 ± 0.084	1.89 ± 0.303	1.93 ± 0.005	3.23 ± 0.006	2.66 ± 0.001
Phenylalanine *	2.08 ± 0.028	2.15 ± 0.049	2.35 ± 0.189	1.83 ± 0.009	2.00 ± 0.003	2.29 ± 0.009	2.38 ± 0.006
Histidine *	1.21 ± 0.042	1.27 ± 0.000	1.55 ± 0.439	0.94 ± 0.001	1.12 ± 0.016	1.42 ± 0.000	1.41 ± 0.000
Methionine *	1.15 ± 0.007	1.16 ± 0.000	1.44 ± 0.004	0.17 ± 0.031	0.44 ± 0.047	1.91 ± 0.581	2.28 ± 0.076
Arginine ***	2.18 ± 0.099	2.45 ± 0.042	3.67 ± 0.063	2.20 ± 0.018	2.55 ± 0.016	3.35 ± 0.001	3.55 ± 0.000
Aspartic acid	3.23 ± 0.028	3.22 ± 0.028	3.68 ± 0.180	2.50 ± 0.012	2.72 ± 0.020	3.16 ± 0.007	3.40 ± 0.005
Glutamic acid	7.94 ± 0.262	8.78 ± 0.014	8.74 ± 0.863	10.01 ± 0.044	10.55 ± 0.036	12.16 ± 0.065	12.39 ± 0.050
Serine	2.03 ± 0.092	2.40 ± 0.141	2.91 ± 0.112	1.75 ± 0.111	2.09 ± 0.006	3.19 ± 0.021	2.93 ± 0.023
Glycine	2.29 ± 0.014	2.65 ± 0.007	4.19 ± 0.832	2.10 ± 0.004	2.84 ± 0.039	4.58 ± 0.042	4.40 ± 0.003
Alanine	2.36 ± 0.014	2.87 ± 0.000	5.28 ± 0.055	2.56 ± 0.009	3.44 ± 0.027	5.82 ± 0.069	5.97 ± 0.001
Cysteine	0.25 ± 0.014	0.35 ± 0.014	1.93 ± 0.957	0.19 ± 0.001	0.28 ± 0.032	0.39 ± 0.077	0.38 ± 0.003
Proline	1.58 ± 0.000	1.52 ± 0.028	2.33 ± 0.124	2.99 ± 0.026	3.60 ± 0.035	4.61 ± 0.044	4.70 ± 0.010
**Total**	**43.49**	**46.62**	**59.72**	**42.56**	**49.39**	**66.34**	**67.02**

* Essential amino acid; ** Conditional essential amino acid; *** Essential amino acid for children.

**Table 2 foods-09-00389-t002:** Fatty acid composition (mg/100 g dry matter basis) of different developmental stages of Italian honey bee drones from Korea (K-) and Buckfast honey bee drones from Denmark (D-).

Fatty Acid	D-Larvae	D-Late Pupae	D-Adult	K-Early Pupae	K-Late Pupae	K-Early Adult	K-Adult
Lauric acid (C12:0)	25.95	31.37	4.08	32.48	33.41	14.17	6.14
Myristic acid (C14:0)	359.51	365.50	15.97	333.07	258.05	48.35	18.31
Palmitic acid (C16:0)	4809.97	4879.12	294.67	4517.45	3570.83	802.94	384.12
Stearic acid (C18:0)	1110.26	1302.45	257.09	1356.94	1267.04	592.54	341.51
Arachidic acid (C20:0)	ND	56.17	35.92	120.62	145.82	157.05	104.24
Behenic acid (C22:0)	ND	ND	62.86	14.38	23.34	51.35	46.46
Lignoceric acid (C24:0)	ND	ND	ND	39.17	42.64	39.99	34.95
**Subtotal (SFA)**	**6305.69**	**6634.61**	**670.59**	**6414.11**	**5341.13**	**1706.39**	**935.73**
Palmitoleic acid (C16:1)	56.35	51.92	166.58	47.65	48.33	74.29	92.91
Elaidic acid (C18:1n9t)	ND	ND	ND	6.75	0.00	0.00	0.00
Oleic acid (C18:1n9c)	4720.25	5104.52	1783.36	4902.83	4412.01	2545.19	1900.32
cis11-Eicosenic acid (C20:1n9)	ND	ND	127.09	8.69	10.38	14.01	9.57
**Subtotal (MUFA)**	**4776.60**	**5156.44**	**2077.03**	**4965.92**	**4470.72**	**2633.49**	**2002.80**
Linoleic acid (C18:2n6c)	ND	67.87	61.8	22.76	30.69	37.43	35.92
Linolenic acid (C18:3n3)	ND	ND	ND	61.24	83.23	108.50	104.85
cis-13,16-Docosadienoic acid (C22:2)	ND	ND	ND	15.20	17.23	21.54	17.56
**Subtotal (PUFA)**	**ND**	**67.87**	**61.8**	**99.20**	**131.15**	**167.47**	**158.33**
**Total**	**11,082.29**	**11,858.92**	**2809.42**	**11,479.23**	**9943.00**	**4507.35**	**3096.86**

**Table 3 foods-09-00389-t003:** Mineral contents (mg/100 g dry matter basis) of different developmental stages of Italian honey bee drones from Korea (K-) and Buckfast honey bee drones from Denmark (D-).

Minerals	D-Larvae	D-Late Pupae	D-Adult	K-Early Pupae	K-Late Pupae	K-Adult
Calcium	34.21	38.7	60.72	43.72	49.29	66.19
Magnesium	68.06	81.86	121.45	82.89	95.03	123.18
Sodium	30.08	38.02	79.45	7.29	8.52	11.33
Potassium	891.08	1101.98	1465.23	544.55	643.06	784.04
Phosphorus	686.88	802.61	1166.06	774.03	892.41	1132.35
Iron	5.62	5.99	12.23	4.86	5.67	10.58
Zinc	5.10	6.04	15.86	5.25	5.88	8.40
Copper	0.11	0.37	1.39	1.82	1.94	2.59
Manganese	0.87	ND	1.71	0.28	0.29	0.52

**Table 4 foods-09-00389-t004:** Antioxidant activities of ethanol extracts of Buckfast honey bee drone larvae, pupae and adults from Denmark. The concentration used for 1,1-diphenyl-2-picryl hydrazyl (DPPH), 2,2-azobis (3-ethylbenzothiazoline-6-sulfonate (ABTS), and the reducing power assay was 500 µg/mL and for the nitrite scavenging assay the concentration was 200 µg/mL. Vitamin C (ascorbic acid) was used as a standard. Different superscripts within a column indicate statistically significant differences (*p* < 0.05).

Extract (mg/mL)	Antioxidant Activity (%)			Reducing Power (700 nm)
	DPPH	ABTS	Nitrite	
Larvae (0.5)	0.3 ± 0.4 ^a^	10.4 ± 0.2 ^a^	25.6 ± 4.5 ^a^	0.018 ± 0.001 ^b^
Late pupae (0.5)	1.3 ± 0.4 ^a^	10.5 ± 0.5 ^a^	20.9 ± 4.1 ^a^	0.008 ± 0.002 ^a^
Adult (0.5)	18.5 ± 1.4 ^b^	40.1 ± 2.3 ^b^	40.4 ± 6.3 ^b^	0.230 ± 0.001 ^c^
Vitamin C (0.1)	92.5 ± 0.6 ^c^	95.2 ± 0.3 ^c^	85.6 ± 2.6 ^c^	1.545 ± 0.064 ^d^

**Table 5 foods-09-00389-t005:** Antimicrobial activities of hot water and ethanol extracts of Buckfast honey bee drone larvae, pupae and adults from Denmark against pathogenic and food spoilage microorganisms. The concentration of extracts and standard chemicals (Ampicillin and Miconazole) used were 500 µg/disc and 1.0 µg/disc, respectively.

Extract	Antimicrobial Activity (Clear Zone: mm)									
	Gram Positive Bacteria				Gram Negative Bacteria				Fungi	
	LM	SE	SA	BS	EC	PA	ST	PV	CA	SC
Larvae	--	--	--	--	--	--	--	--	--	--
Late pupae	--	--	--	--	--	--	--	--	--	--
Adult	--	--	--	--	--	--	--	--	--	--
Ampicillin	13 ± 0.1	21 ± 0.2	15 ± 0.1	12 ± 0.2	6 ± 0.1	8 ± 0.2	11 ± 0.1	18 ± 0.2	--	--
Miconazole	--	--	--	--	--	--	--	--	8 ± 0.1	13 ± 0.2

LM: *Listeria monocytogenes*, SE: *Staphylococcus epidermidis*, SA: *Staphylococcus aureus*, BS: *Bacillus subtilis*, EC: *Escherichia coli*, PA: *Pseudomonas aeruginosa*, ST: *Salmonella typhimurium*, PV: *Proteus vulgaris*, CA: *Candida albicans*, and SC: *Saccharomyces cerevisiae*. ‘—‘indicates no activity.

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
