# Peer review of "Nutritional Composition of Apis mellifera Drones from Korea and Denmark as a Potential Sustainable Alternative Food Source: Comparison Between Developmental Stages"

_foods, 2020, doi:10.3390/foods9040389_

Round 1
Reviewer 1 Report
Dear Authors,
given the attention towards the use of edible insects, your work is current and little investigated.
But it needs important improvements in the description of the materials and methods but above all in the presentation of the results and their discussion.
Below what I think is necessary.
In general re-reading the text, there are some typos.
Abstract
Review the abstract, it does not show that the antioxidant activity has been evaluated only for Buckfast bees.
Introduction
Improve the aim, must be detailed more.
2.1.1. Sample preparation
Specify how many specimens per category and indicate how many replicates have been made.
2.2. Functional properties of Buckfast drone bee ethanol extracts
Explain here or in the results why the evaluation was conducted only on Buckfast drone.
Enter the description of the statistical analysis that has been carried out
Results and discussions
3.1. Body weight of different developmental stages of drone
Buckfast bees? Why aren't the results described?
Figure 1. Detail the statistical analysis in the caption and insert also Buckfast bees.
3.2.1. Amino acid composition
Are there statistically significant differences between the stages and between Italian honey bee drone and Buckfast honey bee drone? As for the composition, one of the two populations disappears from the discussion of the results.
Table 1. Group the columns of the same population.
3.2.2. Fatty acid composition
Delete “Generally, fatty acids, based on the degree of saturation of their acid chain, can be assigned to one of three group: e.g., saturated (SFA), monounsaturated (MUFA) and polyunsaturated fatty acids (PUFA)”.
Authors wrote: “A gradual decreasing trend with developmental stage was found for total fatty acid content”. Is it proven from a statistical point of view? Any differences between the two populations?
Table 2. Group the columns of the same population. Standard deviation? Statistical analysis?
3.2.3. Mineral content
Table 3. Group the columns of the same population. Standard deviation? Statistical analysis?
3.3.1. Total polyphenol, flavonoids, reducing sugar content
Discuss whether the observed values ​​are lower, comparable or better than other foods.
Figure 3. Detail the statistical analysis in the caption.
Conclusions
“On the other hand, the larvae or early pupal stage, with little less protein in comparison to developed stage, can be proposed as animal feed”. How can you talk about the amount of protein if it hasn't been determined?
Rewrite, please.
Author Response
Reviewer 1
Given the attention towards the use of edible insects, your work is current and little investigated.
Thank you.
But it needs important improvements in the description of the materials and methods but above all in the presentation of the results and their discussion.
We have improved the Materials & Methods section and added information on the places of origin of the bees and the way the data were obtained
Below what I think is necessary. In general re-reading the text, there are some typos.
Abstract: Review the abstract, it does not show that the antioxidant activity has been evaluated only for Buckfast bees.
The text was checked for typos and corrected. An explanation why fresh material of the Buckfast bees was not available (which accounts for the fact that some parameters were not available for both populations of bees) is given (see also comment to the Reviewer 1): Buckfast drones arrived freeze-dried from Denmark; fresh Italian drones were available from Korea)
Introduction: Improve the aim, must be detailed more.
We now state: “The usage of drone bees could help enhancing the economic condition especially of the small to medium-scale beekeepers. However, reports of the nutrient composition of honey bee drones are limited. It therefore seemed important to examine and put on record the chemical composition, including amino and fatty acid as well as mineral content of drones to gauge the nutritional value of their different developmental stages. The data would then allow us to draw conclusions on likely functional consequences of drone consumption. The study’s aim was further to clarify if there were any differences in the body composition of honey bee drones from two locations of the northern hemisphere separated by 8,500 km and 20 degrees of latitude.”
2.1.1. Sample preparation: Specify how many specimens per category and indicate how many replicates have been made.
2.2. Functional properties of Buckfast drone bee ethanol extracts: Explain here or in the results why the evaluation was conducted only on Buckfast drone. Enter the description of the statistical analysis that has been carried out
We now write: “Drone larvae, pupae and adults of the honey bee Apis mellifera were obtained from two different countries in two different continents separated by 8,500 km and 20 degrees of latitude (n = 20 for each studied developmental stage of the drones): Buckfast bees from Europe (Denmark) and Italian bees from Asia (South Korea). All analyses were replicated three times.”
“Weights of the different developmental stages of the Italian bee drones were taken with the help of Acculab weighing equipment (ALC310.3) with a precision of 0.001g. However, Buckfast drones were received in a freeze-dried condition from Denmark, which is why their fresh weights could not be recorded.”
“In order to increase reliability, all the analyses were performed in triplicate and expressed as mean ± SD except for the case of fatty acid and mineral analysis. For body weight n=9 individuals were taken for late pupae, early adult and adult Italian bee drone and n=5 individuals were taken for early pupae. The composite sampling technique has been followed for sampling for each experiment. To test the differences, in case of the drone’s weight and the functional properties of Buckfast drones, one-way analysis of variance (ANOVA) followed by Posthoc test were carried out using SPSS 16.1. If the p value (sig.) was found ≤0.05 (CI = 95%), the null hypothesis was rejected.”
Results and discussions: 3.1. Body weight of different developmental stages of drone
Buckfast bees? Why aren't the results described?
As explained above: “Buckfast drones were received in a freeze-dried condition from Denmark, which is why their fresh weights could not be recorded.”
Figure 1. Detail the statistical analysis in the caption and insert also Buckfast bees.
3.2.1. Amino acid composition: Are there statistically significant differences between the stages and between Italian honey bee drone and Buckfast honey bee drone? As for the composition, one of the two populations disappears from the discussion of the results.
The reason for that has been explained (see above). As to the caption of Fig. 1, it now reads: “Body weights of Italian honey bee drones were found to differ significant between pupae and adults, ranging from 0.34 to 0.27g (ANOVA, P<0.05) (Fig. 1).”
Table 1. Group the columns of the same population.
3.2.2. Fatty acid composition: Delete “Generally, fatty acids, based on the degree of saturation of their acid chain, can be assigned to one of three group: e.g., saturated (SFA), monounsaturated (MUFA) and polyunsaturated fatty acids (PUFA)”.
The recommendations were followed.
Authors wrote: “A gradual decreasing trend with developmental stage was found for total fatty acid content”. Is it proven from a statistical point of view? Any differences between the two populations?
Table 2. Group the columns of the same population. Standard deviation? Statistical analysis?
3.2.3. Mineral content :Table 3. Group the columns of the same population. Standard deviation? Statistical analysis?
We did not carry out ant statistical analyses for fatty acid and minerals as these were determined with highly sensitive instruments and methods like GC and ICP-MS. We followed composite sampling techniques in order to avoid heterogeneity of the sample and ran the samples once. Because of one the resultant single value, we cannot carry out statistics.
3.3.1. Total polyphenol, flavonoids, reducing sugar content: Discuss whether the observed values are lower, comparable or better than other foods.
We avoided such a discussion, because the chemicals mentioned are obtained from the plants that the animals feed on and consequently the amounts are always higher in plants than in animals. However, the contents in the plants depend on the species, the seasons and the place from where on animal has ingested the plant (and with it the polyphenols, flavonoids, reducing sugars, etc. Whether the insects want or need any of these compounds based on our current understanding seems meaningless as they cannot help but ingesting these compounds with the plants or plant products they consume. However, we have compared with the values of other edible insects reported in some other publications.
Figure 3. Detail the statistical analysis in the caption.
Such details are now given for all figures and tables (where it was feasible)
Conclusions: “On the other hand, the larvae or early pupal stage, with little less protein in comparison to developed stage, can be proposed as animal feed”. How can you talk about the amount of protein if it hasn't been determined?
The Conclusion has been rewritten in this way: “Similarities in the nutrient content between Italian and Buckfast drones especially with regard to amino and fatty acids were apparent and most likely represent a reflection of the physiological processes shared by the two breeds. On the other hand, discrepancies in the values of some minerals like sodium and potassium in adult populations of both drones very likely stem from distinct environments and their ecological characteristics as well as the different food plants used by the bees. From a nutritional standpoint the relatively small amounts of fatty acids and the richness in minerals would provide an extra advantage. The larvae or the early pupal stage, with less total amino acid and, thus, protein than further developed stages, can be recommended as animal feed. Scientific study showed efficacy of drone brood, although not exact stage mentioned, in terms of improvement of reproductive quality of pigs [83]. However, marketing of drone brood as food per se or a food supplement requires legislation from governmental agencies and permits for the commercialization of this insect. Obviously, a standard dossier of beekeeping practices is essential to ensure the hygiene of the honey bee drone production and to address potential safety issues. Once these obstacles are overcome, the use of drones as food or feed can be a win-win solution for bee keepers and as well as the consumers.”
Reviewer 2 Report
This work could be improved after major revision or complementary information.
The studies made are correct, although not fully novel. The experimental design to assess the nutritional composition of Apis mellifera drones is adequate although more ambitious approach, including the characterization of the polyphenolic profile must be necessary. On the other hand, the study is partially incomplete because not all the parameters have been done for both type of samples (from Denmark and South Korea).
Anyway, the most important objection is related to the recommendation of processing the drones before ingestion. I agree with the authors that based on the haemolysis activity this is the best option, but in this case all the nutritional characterization, or at least that related with thermolabile compounds such as polyphenols, antioxidant properties…, must be done in processed drones.
Author Response
Reviewer 2
The studies made are correct, although not fully novel. The experimental design to assess the nutritional composition of Apis mellifera drones is adequate although more ambitious approach, including the characterization of the polyphenolic profile must be necessary. On the other hand, the study is partially incomplete because not all the parameters have been done for both type of samples (from Denmark and South Korea).
The reviewer is correct with the remark ‘not fully novel’ if s/he thinks of female honey bees, however, no chemical analyses of the kind we conducted (apart from a preliminary one in Korean language) had ever been made on drone bee brood (a to-be-published literature review by Pascal Herren on drone bee publications confirms that). The section on the polyphenolic profile has been rewritten and it is explained that owing to the fact that only freeze-dried, but not fresh material, was available for the sample from Denmark some parameters could not be given for both populations. However, we think it would be unwise not to report the results even if they apply to one or the other population, provided we clearly point this out (which we did).
Anyway, the most important objection is related to the recommendation of processing the drones before ingestion. I agree with the authors that based on the haemolysis activity this is the best option, but in this case all the nutritional characterization, or at least that related with thermolabile compounds such as polyphenols, antioxidant properties…, must be done in processed drones.
The reviewer is correct. However, compounds with antioxidant activity stem from plants, in which they occur in much greater amounts than in the insect and our recommendation to process the fresh drones prior to consumption is to process the insects in ways we have outlined and which the reviewer mentions as “the best option”. One always needs to compromise and we are not recommending that drone bees should be the only food item for humans to accept.
Round 2
Reviewer 1 Report
Dear Authors,
I appreciate the work done but you have not completely responded to what was requested, maybe because I explained myself wrong.
I ask you to make the corrections indicated below.
2.2. Functional properties of Buckfast drone bee ethanol extracts”
Explain here or in the results why the evaluation was conducted only on Buckfast drone.
I ask again why the evaluations were conducted only on Buckfast drone? Specify it.
2.3. Statistical analysis
Specify which posthoc test was used.
Other comments on the statistics follow later.
Figure 1.
Specify with a letter or position the figure corresponding to each population.
Specify the meaning of the letters included in the graphics (posthoc test).
3.2.1. Amino acid composition and Table 1.
You write “(ANOVA, P<0.05 for Italian bee and Buckfast bee).” But haven't you done the posthoc test? Please insert it or specify it.
3.2.2. Fatty acid composition
I suppose the differences are not statistically significant but I ask you to specify it.
3.2.3. Mineral content
I suppose the differences are not statistically significant but I ask you to specify it.
Author Response
We appreciate the review comments. All comments were respected and revised accordingly.
I appreciate the work done but you have not completely responded to what was requested, maybe because I explained myself wrong.
I ask you to make the corrections indicated below.
2.2. Functional properties of Buckfast drone bee ethanol extracts”
Explain here or in the results why the evaluation was conducted only on Buckfast drone.
I ask again why the evaluations were conducted only on Buckfast drone? Specify it.
Answer: Due to limited accessibility of the Italian drone sample we therefore restrict the examination of functional properties for Buckfast bee drone only.
2.3. Statistical analysis
Specify which posthoc test was used.
Answer: LSD was used as posthoc test, and this was clearly stated in MS.
Other comments on the statistics follow later.
Figure 1.
Specify with a letter or position the figure corresponding to each population.
Specify the meaning of the letters included in the graphics (posthoc test).
Answer: Yes, we have included now (Different superscripts indicate statistically significant differences (p<0.05))
3.2.1. Amino acid composition and Table 1.
You write “(ANOVA, P<0.05 for Italian bee and Buckfast bee).” But haven't you done the posthoc test? Please insert it or specify it.
Answer: We only carried out one way ANOVA only. (Although LSD also demonstrated there is significant difference between any two developmental stages of drone population.)
3.2.2. Fatty acid composition
I suppose the differences are not statistically significant but I ask you to specify it.
Answer: Okay. Now we mentioned that this is not examined statistically.
3.2.3. Mineral content
I suppose the differences are not statistically significant but I ask you to specify it.
Answer: Okay. Now we mentioned that this is not examined statistically.
Reviewer 2 Report
Some questions related to the characterization of the nutritional profile in processed drones are still at the same point. However, considering that the paper could understand it as an initial paper proposing a new food item for humans, after the correction of the manuscript and the authors’ replies, the manuscript is fine to be accepted for publication in the present form.
Author Response
we appreciate the effort to make our manuscript more meaningful and clear.
Thank youi